# Photosynthetic, Respirational, and Growth Responses of Six Benthic Diatoms from the Antarctic Peninsula as Functions of Salinity and Temperature Variations

**DOI:** 10.3390/genes13071264

**Published:** 2022-07-16

**Authors:** Lara R. Prelle, Ina Schmidt, Katherina Schimani, Jonas Zimmermann, Nelida Abarca, Oliver Skibbe, Desiree Juchem, Ulf Karsten

**Affiliations:** 1Applied Ecology and Phycology, Institute of Biological Sciences, Albert-Einstein-Strasse 3, University of Rostock, 18057 Rostock, Germany; lara.prelle@uni-rostock.de (L.R.P.); ischmidt@uni-bremen.de (I.S.); desiree.juchem@uni-rostock.de (D.J.); 2Botanic Garden and Botanical Museum Berlin-Dahlem, Freie Universität Berlin, 14163 Berlin, Germany; k.schimani@bo.berlin (K.S.); j.zimmermann@bo.berlin (J.Z.); n.abarca@bo.berlin (N.A.); o.skibbe@bo.berlin (O.S.)

**Keywords:** growth rate, climate change, tolerance, ecophysiology, 18 S, *rbc*L

## Abstract

Temperature and salinity are some of the most influential abiotic parameters shaping biota in aquatic ecosystems. In recent decades, climate change has had a crucial impact on both factors—especially around the Antarctic Peninsula—with increasing air and water temperature leading to glacial melting and the accompanying freshwater increase in coastal areas. Antarctic soft and hard bottoms are typically inhabited by microphytobenthic communities, which are often dominated by benthic diatoms. Their physiology and primary production are assumed to be negatively affected by increased temperatures and lower salinity. In this study, six representative benthic diatom strains were isolated from different aquatic habitats at King George Island, Antarctic Peninsula, and comprehensively identified based on molecular markers and morphological traits. Photosynthesis, respiration, and growth response patterns were investigated as functions of varying light availability, temperature, and salinity. Photosynthesis–irradiance curve measurements pointed to low light requirements, as light-saturated photosynthesis was reached at <70 µmol photons m^−2^ s^−1^. The marine isolates exhibited the highest effective quantum yield between 25 and 45 S_A_ (absolute salinity), but also tolerance to lower and higher salinities at 1 S_A_ and 55 S_A_, respectively, and in a few cases even <100 S_A_. In contrast, the limnic isolates showed the highest effective quantum yield at salinities ranging from 1 S_A_ to 20 S_A_. Almost all isolates exhibited high effective quantum yields between 1.5 °C and 25 °C, pointing to a broad temperature tolerance, which was supported by measurements of the short-term temperature-dependent photosynthesis. All studied Antarctic benthic diatoms showed activity patterns over a broader environmental range than they usually experience in situ. Therefore, it is likely that their high ecophysiological plasticity represents an important trait to cope with climate change in the Antarctic Peninsula.

## 1. Introduction

Global warming is unequivocal, as is now evident from observations of increases in average global air and ocean temperatures, leading to the widespread melting of snow and ice in the polar regions, and rising average global sea levels [1]. However, the effects are quite different in Antarctica and the Arctic. While global warming is already strongly affecting the whole Arctic region [2], in Antarctica, thus far, mainly the Antarctic Peninsula has gotten warmer, where the air temperature and near-surface sea temperature have risen by 3 °C and 1 °C, respectively, in the past 50 years. This has resulted in a significant retreat of ice shelves, increased coastal erosion, less snow, and more meltwater and rain, with strong consequences for Antarctic organisms and ecosystems [1,3].

A particularly ecologically important group of eukaryotic microorganisms in Antarctic and Arctic shallow-water coastal zones are benthic diatoms (living on top of or associated with sediments or rocks), which are poorly studied in terms of biodiversity, biogeography, and ecology. Knowledge on these phototrophs, which form a key assemblage known as microphytobenthos, stems mainly from temperate to tropical marine soft-bottom regions worldwide—for example, tidal flats. Here, they exert multiple important functions as high primary producers providing a major food source for benthic suspension- or deposit-feeders [4], as control barriers for oxygen/nutrient fluxes at the sediment–water interface [5], and as stabilizers of sediment surfaces against hydrodynamic erosion by the excretion of sticky extracellular polymeric substances [6]. Microphytobenthic communities, together with planktonic diatoms, contribute about 45% of total marine carbon fixation [7].

A pioneer study of the Young Sound, Greenland, indicated that benthic diatoms contributed to 40% of the total marine primary production (60% originated from kelps) [8]. These data were later confirmed for the Arctic Kongsfjorden (Svalbard, Norway), in which microphytobenthic production was as high as in temperate regions [9,10,11]. Consequently, benthic diatoms are regarded as playing an exceptionally important role in Arctic coastal food webs [12], while similar studies for Antarctica are lacking. To this day, remarkably little is known about marine benthic diatom biodiversity and ecophysiology in Antarctica. In contrast, the various microalgae of the Antarctic phytoplankton, as well as those associated with sea ice, with snow fields, or inhabiting terrestrial sites, are much better studied ([13,14,15,16], and references therein). In addition, more recent publications have applied next-generation sequencing (NGS) technologies, greatly expanding current knowledge by providing fundamental information on the underlying molecular mechanisms of physiological and biochemical adaptations to polar environmental conditions.

The Antarctic microphytobenthos—especially along the Antarctic Peninsula—experiences strong seasonal variability in abiotic parameters on the edge of extremes such as temperature, salinity, and light availability (e.g., polar day versus polar night). Sea ice coverage has a strong effect on photosynthesis, as light penetration decreases with ice thickness and snow coverage [17,18]. However, coastal regions of the Antarctic Peninsula face strong deglaciation due to warming and, hence, are less and less covered with ice, allowing benthic communities to occupy and develop on such pristine sediments [19]. The temperature variations in this region can lead to freezing in winter and melting of snow and ice during the warm season. Especially shallow coastal sites are strongly influenced by melting and freezing processes, along with temperature, creating strong seasonal salinity changes. Increasing melting enhances terrestrial freshwater runoff, thereby decreasing salinity, and vice versa—freezing in winter decreases freshwater runoff, increasing salinity in the remaining aquatic environment. In addition, the Antarctic Peninsula receives 450 mm of precipitation per year; however, due to global warming, this is now mainly received as rain [20], thereby also decreasing the sea surface salinity.

The few data available indicate that Antarctic benthic diatoms generally live most of the time under low-light conditions. Nevertheless, these phototrophic microorganisms have been reported to adjust their photosynthetic activity very efficiently to changing irradiance [21,22]. The benthic diatom *Trachyneis aspera* was found to grow at ambient photon fluence rates of <1 µmol photons m^−2^ s^−1^, with saturated photosynthetic rates (I_k_ values) already between 7 and 16 µmol photons m^−2^ s^−1^ [23]. Hence, benthic diatoms, in virtue of their low light requirements for photosynthesis, are capable to colonize deeper soft bottoms [4]. The ability of Antarctic benthic diatoms to acclimate not only to such extreme low-radiation environments, but also to high-radiation environments, has been documented in some studies [21,22]. Two underlying processes for the regulation of the rapid switch from a light-harvesting to a photoprotective state have been reported: One is non-photochemical fluorescence quenching—a mechanism involving the quenching of singlet excited-state chlorophylls via enhanced internal conversion back to the ground state of these pigments. As a consequence, excessively absorbed radiation energy is harmlessly dissipated as heat through molecular vibrations [24]. The second process is the cycling of electrons around photosystem II and/or photosystem I [24]. Both mechanisms support the safe dissipation of excessively absorbed radiation energy during a sudden increase in the incident light conditions, and contribute to a rather unusual photosynthetic flexibility in diatoms, providing optimal photoprotection and rapid photoacclimation under the fluctuating and highly variable Antarctic light climate. In addition, many but not all benthic diatoms also exhibit a behavioral trait in response to changes in the light field by vertical migration into or out of the sediment to avoid photoinhibition [25].

While Arctic benthic diatoms can be characterized as eurythermal and psychrotolerant microalgae with growth optima at around 15 °C [26,27], this seems to be in sharp contrast to their Antarctic counterparts, which show stenothermal and psychrophilic traits [28]. Psychrophilic and psychrotolerant species can be physiologically distinguished, as the former can survive at freezing temperatures but will die at more moderate temperatures [29]. Typical examples include the psychrotolerant green microalga *Coccomyxa subellipsoidea* C-169, which was isolated from a terrestrial site in Antarctica [30], and the psychrophilic unicellular green alga *Chlamydomonas* sp. ICE-L that thrives in floating Antarctic sea ice [14]. Psychrophilic traits are exemplarily documented in the Antarctic benthic diatoms *Odontella litigiosa* and *Gyrosigma subsalinum* var. *antarctica*, both collected at Potter Cove, King George Island, which exhibit optimal growth at 0 °C and full inhibition of cell division at <7–9 °C [28]. Whether other Antarctic benthic diatoms follow the low temperature demand for growth is unknown, but already Wiencke and Tom Dieck [31] have reported extremely low temperature requirements for growth and survival in various seaweed species endemic to Antarctica, compared to the Arctic and more temperate regions. Although the number of such ecophysiological studies is small, it can be assumed that the conspicuous differences in the temperature requirements for growth in Arctic and Antarctic benthic diatoms are related to the much longer isolation and cold-water history of the Southern polar region (at least 23 million years [32]) compared to the Northern high latitudes (approximately 2 million years). These striking differences in both cold-water systems have supported the development of many endemic marine organisms in Antarctica, most of which are extremely sensitive to warming [33].

In contrast to the fragmentary data on marine benthic diatoms, Antarctic freshwater diatoms are much better studied in terms of biodiversity, ecology, and biogeography [34]. These authors studied biogeographic patterns of freshwater diatom communities of >400 lakes spread across the Antarctic realm, and identified highly distinct diatom florae, in terms of both composition and richness. More importantly, 44% of all determined species were convincingly reported to be endemic to Antarctica [34].

In contrast to light and temperature conditions, salinity is typically a local environmental factor, which may strongly vary in Antarctic near-shore waters, where meltwater—particularly during summer—mixes with marine water bodies. Here, horizontal and vertical gradients between freshwater and fully marine conditions can be measured. In addition, tidal flows, hydrological conditions, wind, precipitation, and evaporation strongly influence the salt concentration of the water body in question. The effect of salinity on benthic diatoms and other algae from polar waters is generally little studied, in strong contrast to temperate regions [35,36].

In this study we investigated the ecophysiological response patterns of six Antarctic benthic diatom strains under a temperature, light, and salinity gradient. The diatoms were isolated from two marine shallow-water stations and one freshwater reservoir at Potter Cove, Antarctic Peninsula, and subsequently identified using morphological and molecular markers. Based on the study of Longhi et al. [28], we expected stenothermal and low-light acclimated response patterns in terms of photosynthesis and growth. In addition, we expected euryhaline and stenohaline responses for the marine and freshwater isolates, respectively.

## 2. Materials and Methods

### 2.1. Study Site

Sediment surface samples taken in January/February 2020 from four study sites (Figure 1, Table 1) near the Argentinian research station Carlini Base (S 62°14′17.45″, W 58°40′2.19″) on King George Island were used for benthic diatom isolation. All isolates were established from samples collected in Potter Cove, which is separated into an inner part with a maximum depth of 50 m, and an outer part connected to the open ocean, with maximum depths of 100–200 m [37].

### 2.2. Culture Establishment and Maintenance Conditions

Six unialgal benthic diatom strains were established: *Navicula criophiliforma* A. Witkowski, C. Riaux-Gobin, and G. Daniszewska-Kowalczyk (Naviculaceae, strain APC06 D288_003), *Chamaepinnularia gerlachei* Van de Vijver and Sterken (Naviculaceae, strain APC14 D296_001), *Navicula concordia* C. Riaux-Gobin and A. Witkowski (Naviculaceae, strain APC28 D310_004), *Nitzschia annewillemsiana* Hamsher, Kopalová, Kociolek, Zidarova, and Van de Vijver (Bacillariaceae, strain APC18 D300_012), *Planothidium* sp. (Achnanthidiaceae, strain APC18 D300_015), and *Psammothidium papilio* (D.E. Kellogg, Stuiver, T.B. Kellogg, and G.H. Denton) K. Kopalová and B. Van de Vijver (Achnanthaceae, strain APC18 D399_023).

The marine culture *N. criophiliforma* originated from sample location APC06 (S 62°14′30.55″, W 58°40′54.96″), which was an intertidal rock pool. Due to its location in the intertidal zone, abiotic parameters such as temperature and salinity strongly varied. *C. gerlachei* was isolated from a sample at the inner part of Potter Cove (APC14, S 62°13′43.61″, W 58°39′49.36″), at 15 m depth, from a biofilm on top of a sediment. Unfortunately, oxidized material from the strain *N. criophiliforma* had low quality, and species identification on this isolate alone was not possible. A genetically identical species was isolated from brackish water at sample location APC12, and material of this strain was used for identification. *Navicula concordia* from a sample location at the outer cove (S 62°14′16.50″, W 58°42′44.20″), at 5 m depth, originated from a biofilm on top of stones. According to Hernández et al. [38], the minimum water temperature was measured at −1.69 °C in the inner part of Potter Cove and −1.4 °C at the outer part, while the maximum temperature was 2.89 °C and 1.98 °C, respectively. Furthermore, the salinity of the outer cove is stable at ca. 33.5 S_A_, while the salinity in the inner cove can drop down to 29.6 S_A_.

The limnic isolates (*N. annewillemsiana*, *Planothidium* sp., and *P. papilio*) were established from biofilms on top of stones in a freshwater drinking reservoir (S 62°14′16.30″, W 58°39′44.10″). During sampling, no measurements of pH, temperature, or conductivity were taken, due to malfunctioning instruments.

The diatom cells were isolated from aliquots of environmental samples to establish clonal cultures. Under an inverse light microscope (100–400× magnification, Olympus, Japan), single cells were transferred using a microcapillary glass pipette onto microwell plates containing culture medium (Guillard’s f/2 medium [39,40] or Walne’s medium [41]; 34 S_A_ for marine samples and 1 S_A_ for freshwater samples). All samples and isolated diatom cells from Antarctica were maintained at 5–7 °C. Irradiation was provided by white-light LEDs at 5000 K under a 16:8 h light:dark cycle, with several dark phases during the day to prevent photo-oxidative stress. After successful establishment of clonal cultures, they were separated into subsamples for DNA extraction, morphological analysis, and ecophysiological experiments. For the latter, diatom cultures were cultivated in sterile filtered Baltic Sea water, enriched with Guillard’s f/2 medium [39,40] and sodium metasilicate (Na_2_SiO_3_ • 5 H_2_O; 10 g 100 mL^−1^) to a final concentration of 0.6 mM (further referred to as cultivation medium). Salinity of 33 S_A_ for the marine cultures was adjusted by using artificial sea salt (hw-Marinemix^®^ professional, Wiegandt GmbH, Germany), while 1 S_A_ for the limnic cultures was achieved by dilution with deionized water.

All stock cultures for the ecophysiological experiments were kept at 8–9 °C at 15–20 μmol photons m^−2^ s^−1^ under a 16:8 h light:dark cycle (Osram Daylight Lumilux Cool White lamps L36W/840, Osram, Munich, Germany).

### 2.3. Acquisition and Identification of Morphometric Data

In order to obtain clean diatom frustules, material harvested from the unialgal cultures was treated with 35% hydrogen peroxide at room temperature to oxidize the organic material, and then washed with distilled water. To prepare permanent slides for light microscopy (LM) analyses, the cleaned material (frustules and valves) was dispersed on glass coverslips, dried at room temperature, and embedded with the high-refraction-index mounting medium Naphrax^®^ (Morphisto GmbH, Offenbach, Germany).

Observations were conducted with a Zeiss Axioplan Microscope equipped with differential interference contrast (DIC), using a Zeiss 100 × Plan Apochromat objective, and microphotographs were taken with an AXIOAM MRc camera. Aliquots of cleaned sample material for scanning electron microscopy (SEM) observations were mounted on stubs and observed under a Hitachi FE 8010 scanning electron microscope operated at 1.0 kV.

### 2.4. DNA Extraction, Amplification, and Sequencing

Culture material was transferred to 1.5 mL tubes. DNA was isolated using the NucleoSpin^®^ Plant II Mini Kit (Macherey and Nagel, Düren, Germany), following the manufacturer’s instructions. DNA fragment size and concentrations were evaluated via gel electrophoresis (1.5% agarose gel) and NanoDrop^®^ (PeqLab Biotechnology LLC; Erlangen, Germany), respectively. DNA samples were stored at −20 °C until further use, and finally deposited in the Berlin collection of the DNA Bank Network [42].

Amplification was conducted by polymerase chain reaction (PCR) as described by Zimmermann et al. [43] for the V4 region of 18 S. For the strain *C. gerlachei*, the whole 18 S gene was amplified as described by Jahn et al. [44]. The protein-coding plastid gene *rbc*L was amplified as described by Abarca et al. [45]. PCR products were visualized in a 1.5% agarose gel and cleaned with MSB Spin PCRapace^®^ (Invitek Molecular GmbH; Berlin, Germany), following the manufacturer’s instructions. The samples were normalized to a total DNA content > 100 ng/µL using NanoDrop (PeqLab Biotechnology) for further sequencing. Sanger sequencing of the PCR products was conducted bidirectionally by Starseq^®^ (GENterprise LLC; Mainz, Germany).

### 2.5. Data Curation

Vouchers and DNA of all strains were deposited in the collections at Botanischer Garten und Botanisches Museum Berlin, Freie Universität Berlin (B). DNA samples were stored in the Berlin DNA bank, and are available via the Genome Biodiversity Network (GGBN) [46]. All sequences were submitted to the European Nucleotide Archive (ENA, http://www.ebi.ac.uk/ena/) using the software tool annonex2embl (accessed on 11 July 2022) [47] and can be retrieved from ENA under the study number PRJEB54671. All cultures are available from the authors at the culture collection of the Department of Applied Ecology and Phycology, University of Rostock.

### 2.6. Photosynthetic Efficiency

The photosynthetic potential of the six Antarctic benthic diatom strains as a function of salinity and temperature was measured using the pulse amplitude modulation (PAM) approach (PAM-2500, Heinz Walz GmbH, Effeltrich, Germany). The effective photochemical quantum yield of photosystem II in light-adapted cells, Y(II), was calculated (Equation (1)) by measurement of Fm′ (maximum chlorophyll fluorescence yield) and F (base fluorescence):(1)$$Y\left(\mathrm{II}\right)=\frac{\left(Fm^{\prime }-F\right)}{{\mathrm{Fm}}^{\prime }}$$

Equation (1): Calculation of the effective photochemical quantum yield of photosystem II (Y(II))

Intensity of the measured light and gain were adjusted to reach F_t_ (continuous base fluorescence) values between 0.15 and 0.2. Measurements were excluded from calculation when biomass did not surpass the F_t_ value of 0.15 at the highest measured light intensity and gain.

All cultures were kept under culture conditions before transfer into the respective test media, with three replicates of. Two drops of thin diatom culture suspension were applied on a 25 mm glass-fiber filter (GF/6, Whatman, Little Chalfont, UK) and incubated in 2 mL of the respective treatment medium. To avoid nutrient deficiency, 1 mL of the medium was replaced with fresh medium every day. A radiator block was used during the measurements to avoid excessive temperature stress in the laboratory.

Different salinity treatments were performed using sterile, filtered deionized water and artificial sea salt, with the addition of cultivation media. For the salinity treatments, marine isolates were incubated for three days under cultivation conditions at 1, 5, 10, 15, 25, 35, 45, 55, 65, 75, 85, and 100 S_A_. Limnic isolates were exposed to salinities of 1, 5, 10, 20, 30, 35, 40, 45, 55, and 65 S_A_. The isolates were incubated for three days prior to PAM measurements.

For the temperature treatments, experimental media were based on sterile, filtered deionized water and artificial sea salt (1 S_A_ for limnic cultures and 35 S_A_ for marine cultures), with the addition of cultivation media. All isolates were incubated for five days (t_5_) at average temperatures of 1.5, 5, 7, 10, 15, 20, and 25 °C. Temperatures were achieved using a temperature organ. For 1.5 °C, an ice bath was used, with an exchange of ice every 12 h. The isolates were incubated for three days prior to PAM measurements.

PAM measurements were performed every 24 h, starting at day 0 (t_0_), immediately after the transfer of the diatom cells onto the filter, until t_3_ and t_5_.

### 2.7. Light Irradiance Curves (P–I Curves)

Photosynthetic activity as a function of light availability was measured as described by Prelle et al. [48] in a self-constructed P–I (photosynthesis–irradiance) box. Three (*n* = 3) airtight oxygen electrode chambers (DW1, Hansatech Instruments, King’s Lynn, UK), tempered at 8 °C, were filled with 3 mL of thin algal-log-phase suspension, with the addition of 30 μL of sodium bicarbonate (NaHCO_3_, final concentration 2 mM) to avoid carbon deficit during the measurements. Oxygen concentration measurements along 10 increasing photon flux density levels, ranging from 3.6 to ~1670 μmol photons m^−2^ s^−1^ of photosynthetically active radiation (PAR), were undertaken using oxygen dipping probe DP sensors (PreSens Precision Sensing GmbH, Regensburg, Germany) linked to fiber-optic oxygen transmitters via optical fibers (Oxy 4 mini meter, PreSens Precision Sensing GmbH, Regensburg, Germany). Measurements started with a 30-min respirational phase, followed by a series of 10-min photosynthetic phases for each increasing light level.

Chlorophyll *a* concentration per chamber was measured after each final measurement by the extraction of 3 mL of the algal suspension using 96% ethanol (*v*/*v*). Chlorophyll *a* was measured spectrophotometrically at 665 nm and 750 nm (Shimadzu UV-2401 PC, Kyoto, Japan) [49], and calculated according to Equation (2):(2)$$µg\;Chl\;a=\frac{\left(E665-E750\right)xvx10^{6}}{83xVxd}$$

Equation (2): Chlorophyll *a* calculation, where v is the extraction volume (mL), d is the cell length (cm), and V is the volume of filtered suspension (mL).

Due to photoinhibition in some of the diatom strains, the mathematical photosynthesis model of Walsby [50] was used for fitting and calculation of the maximum rates of net primary production (NPP_max_), respiration (R), light utilization coefficient (α), photoinhibition coefficient (β), light saturation point (I_k_), and light compensation point (I_c_).

### 2.8. Temperature-Dependent Photosynthesis and Respiration

Photosynthetic and respirational rates of the six Antarctic diatom strains in response to temperatures between 5 °C and 40 °C were measured using the same P–I box as for the P–I curves, following the approach of Karsten et al. [51]. In contrast to the P–I curves, light was switched off during the respirational phase and set to photosynthesis-saturated 342.2 ± 40 μmol photons m^−2^ s^−1^ PAR during the photosynthetic phase. Starting at 5 °C, a 20-min dark incubation phase was followed by a 10-min respirational phase and a 10-min photosynthetic phase. Afterwards, the temperature was increased by 5 °C, and the process was repeated until reaching 40 °C. Oxygen concentration measurements were also normalized to the total Chlorophyll *a* concentration [49].

### 2.9. Growth Rates

Growth rates of the marine diatom strain *C. gerlachei* and the limnic diatom strain *P. papilio* in response to salinity and temperature were determined as described by Karsten et al. [52], Gustavs et al. [53], and Prelle et al. [54]. Measurement of the in vivo fluorescence of chlorophyll *a* was used as a proxy for biomass. Using an MFMS fluorimeter (Hansatech Instruments, King’s Lynn, UK), Chlorophyll *a* was excited by blue light emission and detected as relative units by an amplified photodiode that was separated from scattered excitation light. This method is particularly suitable for benthic diatoms, as chlorophyll *a* fluorescence units correlate well with chlorophyll *a* and cell carbon concentrations, as well as cell numbers, in diatoms [52,53]. Both diatom cultures were cultivated in disposable Petri dishes (*n* = 3) with cover lids, in a volume of 15 mL of the respective treatment medium, and measured every 24 h for 8 days. To avoid the measurement of potential initial shock reactions of the isolates, 1 mL of log-phase algae suspension was incubated in 14.5 mL of the respective trial medium for four days under experimental conditions.

Growth as a function of salinity was tested by exposure to salinities of 1, 5, 25, 35, 45, 65, 85, and 100 S_A_ for the marine species *C. gerlachei*, and 1, 5, 10, 20, and 30 S_A_ for the limnic species *P. papilio*. Salinities were adjusted using artificial sea salt (hw-Marinemix^®^ professional) dissolved in deionized water. All cultures were enriched with f/2 and metasilicate, and kept at 8–9 °C under standard cultivation conditions.

Furthermore, growth in response to five temperatures (5, 8, 15, 20, and 30 °C) at salinities of 1 S_A_ (*C. gerlachei*) and 35 S_A_ (*P. papilio*), with added f/2 and metasilicate, was investigated. Treatments at 5 °C were kept in a wine storage refrigerator with added LEDs; treatments at 8, 15, and 20 °C were kept in climate chambers; and treatments at 30 °C were carried out in a tempered water bath, with all reflecting light settings similar to cultivation conditions. After measurements, the growth rates of the logarithmic phase were calculated using the following equation: N = N_0_ × e^(µ × dt)^, where N is the fluorescence on the measuring day, dt is the difference in time (days) between the measuring day and the starting day, and µ is the growth rate) [53].

### 2.10. Statistical Analysis

Our statistical analysis was similar to that of Prelle et al. [54], as Microsoft Office Excel (2016) was used for the calculation—partially by application of the solver function, by minimizing the sum of normalized squared deviations for the fitting of the model of Walsby [50]—and creation of figures. R (Version: 4.0.2) was used for the calculation of significance levels using one-way ANOVA followed by a post hoc Tukey’s honestly significant differences test (critical *p*-value < 0.05), as well as for the fitting of the model of Yan and Hunt [55] for the temperature-dependent photosynthesis. Confidence intervals were calculated using the library nlstools in R.

## 3. Results

### 3.1. Species Identification

Five of the Antarctic isolates were identified to the species level, and one to the genus level. Table 2 lists the taxa, with information on the morphology and respective accession numbers of the marker genes *rbc*L and 18 SV4/18 S, while Figure 2, Figure 3 and Figure 4 depict the LM and SEM images.

APC14 D296_001 was identified as *Chamaepinnularia gerlachei* Van de Vijver and Sterken (Figure 2L–T; valves of strain D294_006 are depicted as well, since this strain was used as a supplement for identification). This species was first published in the work of Van de Vijver et al. [56], from dry soil samples collected at James Ross Island, near the northeastern extremity of the Antarctic Peninsula, and has been observed only in maritime Antarctica thus far [57,58,59].

*Navicula concordia* (Figure 3A–K) was identified as *N. concordia* C. Riaux-Gobin and A. Witkowski, and APC06 D288_003 as *Navicula criophiliforma* A. Witkowski, C. Riaux-Gobin, and G. Daniszewska-Kowalczyk (Figure 2A–K). Both were first published in the work of Witkowski et al. [60], from the Kerguelen Islands coastal area, in the Southern Ocean. Recently, *N. criophiliforma* was reported from Livingston Island, north of the Antarctic Peninsula [61]. This species formed auxospores during cultivation, leading to high variance in the dimensions of the valves.

APC18 D300_012 (Figure 3L–U) was identified as *Nitzschia annewillemsiana* Hamsher, Kopalová, Kociolek, Zidarova, and Van de Vijver. It was first published in the work of Hamsher et al. [62], from freshwater and wet soil samples from James Ross Island and the South Shetland Islands, and has been only reported from this area to date [59].

APC18 D300_023 was identified as *Psammothidium papilio* (D.E. Kellogg, Stuiver, T.B. Kellogg, and G.H. Denton) K. Kopalová and B. Van de Vijver (Figure 4M–Z). It was first described as *Navicula papilio* by Kellogg et al. [63], but this species has been reported several times from maritime Antarctica under different synonyms [32,57,59,64].

APC18 D300_015 was identified to the genus level as *Planothidium* sp. (Figure 4A–L). There was a high morphological resemblance to *P. frequentissimum* (Lange-Bertalot) Lange-Bertalot. However, molecular data showed differences in both marker genes compared to *P. frequentissimum* strains from GenBank. There were 4 base-pair differences in 18 SV4 and 20 in the *rbc*L gene compared to the *P. frequentissimum* strain PF1 (Accession numbers: KJ658409 and KJ658392). In comparison to the strain D06_139 (Accession numbers: KY650786 and KX650815), 11 bp differences were found in 18 SV4, and 19 in *rbc*L.

### 3.2. Photosynthetic Potential

The photosynthetic potential of all six diatom strains exhibited wide tolerance ranges between the tested salinities from 1 S_A_ to 100 S_A_ after three days of incubation (Figure 5, Appendix A). The overall highest and lowest optimal quantum yields were measured for *N. concordia*, with 0.595 at 45 S_A_ and 0.033 at 5 S_A_, respectively. The three marine species *N. criophiliforma*, *C. gerlachei*, and *N. concordia* exhibited typical tolerance curve patterns, with significant optima at 25–35 S_A_, 25–45 S_A_ and 5–45 S_A_, respectively (*p* < 0.05, Figure 5). The tolerance range of *N. criophiliforma* was narrower compared to both other marine species after calculation of the range of the highest effective quantum yield at the 80th percentile and above, between the 20th and 80th percentiles, and below the 20th percentile (Figure 6A). This taxon exhibited high effective quantum yields (upper 80th percentile) at only two experimental salinities, while the other isolates covered 4–6 salinities (Figure 6A). Nevertheless, all marine species exhibited a moderate effective quantum yield (between the 20th and 80th percentiles), ranging from 10 to 55/75 S_A_ and up to 100 S_A_. In contrast, the two limnic species *Planothidium* sp. and *P. papilio* showed the highest significant optima at 10–20 S_A_ and 1–10 S_A_, respectively (*p* < 0.05, Figure 5). Salinities higher than 10/20 S_A_ resulted in a decreasing effective quantum yield up to 40/55 S_A_. Due to low biomass, *Planothidium* sp. was only tested in two salinities, of which the highest effective quantum yield was measured at 10 S_A_.

The photosynthetic potential of the six diatom strains after five days of incubation also exhibited broad tolerances for the investigated temperature range of 1.5 to 25 °C (Figure 7, Appendix A). The highest overall effective quantum yield was found for *N. concordia*, with 0.585 at 15 °C, while the lowest was for *Planothidium* sp., with 0.123 at 25 °C. Between 1.5 and 25 °C, only small significant deviations of the effective quantum yield were found for all three marine taxa, as well as for *P. papilio*, while for *N. annewillemsiana* and *Planothidium* sp. The highest effective quantum yield was at 15 to 25 °C (*p* < 0.05, Figure 7). With the exception of *N. criophiliforma* at 25 °C and *C. gerlachei* at 1 °C and 25 °C, all species exhibited moderate photosynthetic potential between 1.5 °C and 25 °C. In comparison to t_5_, significance levels of t_0_ between each temperature treatment of the marine species were not as distinct as for the limnic species (Figure 7).

### 3.3. Light-Dependent Photosynthesis

The photosynthetic and respirational rates of all six diatom strains exhibited species-specific responses towards increasing photon fluence rates, resulting in different P–I parameters (Figure 8, Table 3). The overall highest NPP_max_ was for the marine species *N. criophiliforma*, with 202.3 ± 45.4 µmol O_2_ mg^−1^ Chl *a* h^−1^, which was at least twice as high as that of the remaining isolates. Respiration rates varied among the isolates, between −47 ± 8.9 µmol O_2_ mg^−1^ Chl *a* h^−1^ (*N. criophiliforma*) and −10.5 ± 3.1 µmol O_2_ mg^−1^ Chl *a* h^−1^ (*N. concordia*) (Figure 8, Table 3). All isolates had low light compensation points (I_c_), varying significantly between 5.8 ± 1 μmol photons m^−2^ s^−1^ (*N. concordia*) and 17.5 ± 3 μmol photons m^−2^ s^−1^ (*Planothidium* sp.) (*p* < 0.05, Table 3). The light saturation points (I_k_) for all six isolates ranged between 64 ± 11.5 μmol photons m^−2^ s^−1^ (*N. criophiliforma*) and 16.3 ± 3.9 μmol photons m^−2^ s^−1^ (*N. annewillemsiana*). Photoinhibition was detected in almost all isolates except for *Planothidium* sp. and *P. papilio*. The highest photoinhibition was found in *N. criophiliforma*, with −0.03 ± 0.02, which was, however, not significant between *C. gerlachei*, *N. concordia*, and *N. annewillemsiana* (*p* < 0.05, Table 3).

### 3.4. Temperature-Dependent Photosynthesis and Respiration

Photosynthetic and respirational responses under increasing temperatures from 5 to 40 °C resulted in individual response patterns (Figure 9, Appendix A). Photosynthesis and respiration rates typically rose with increasing temperature and decreased after reaching the optimal temperature. The overall highest photosynthesis and respiration rates were measured for *N. criophiliforma* at 15 °C and 30 °C, respectively (Figure 9). In general, positive net photosynthetic rates ranged between 5 °C and 25/30 °C, with varying optima for each strain (from 5 °C to 20 °C). At temperatures > 25/30 °C, photosynthesis was inhibited, and only respirational oxygen consumption could be measured (Figure 9). Respirational rates could be detected over the entire temperature range from 5 to 40 °C, with optima between 20 and 35 °C (Figure 9). Fitting of the measured data using the model of Yan and Hunt [55] revealed maximum photosynthetic rates of the marine isolates between 11.1 and 15.7 °C, and a positive net photosynthesis up to 32.5 and 35.6 °C, respectively (Table 4). The optimal temperature for the limnic species was slightly lower—between 3.0 and 12.5 °C, with positive net photosynthesis up to 26.0 and 33.5 °C. Fitting of the respirational data resulted in much higher optimal temperatures, ranging between 26.6 °C (*N. annewillemsiana*) and 30.6 °C (*C. gerlachei*), with maximal values up to 44.4 °C (*N. concordia*).

### 3.5. Growth Rates

One marine and one limnic culture were exemplarily investigated for growth as a function of salinity and temperature (Figure 10, Appendix A). *C. gerlachei* exhibited a strong optimum at 15 °C, with growth rates of 0.84 µ d^−1^, while the optimal growth temperature for *P. papilio* ranged between 5 and 15 °C, with similar growth rates around 0.4 µ d^−1^ (Figure 10). Both diatom strains were unable to grow at temperatures > 20 °C. Using the model of Yan and Hunt [55], the optimal growth temperature of >80% of the maximal growth ranged from 6.5 to 19.9 °C for *C. gerlachei*, and from 1.6 to 14.5 °C for *P. papilio* (Table 4). The overall maximal growth rate for *C. gerlachei* was at 13.0 °C (0.44 µ d^−1^), and for *P. papilio* it was at 6.5 °C (0.30 µ d^−1^). Growth rates as a function of salinity for the marine species *C. gerlachei* were determined over a range from 1 to 65 S_A_. This species exhibited a broad salinity tolerance, as reflected in growth rates between 0 and 79.2 S_A_ (0.2 to 0.4 µ d^−1^), with a distinct optimum at 6.5 S_A_ (0.58 µ d^−1^) (Table 4). In contrast, the limnic species *P. papilio* grew only over a range of 1 to 20 S_A_, with optima between 1 and 10 S_A_. The model calculation for salinity exhibited highest growth rate of 0.42 µ d^−1^ at 5.28 S_A_ (Table 4). The optimal growth range >80% growth rate for this isolate ranged between 0.9 and 13.7 S_A_.

## 4. Discussion

All six marine and limnic benthic diatom species from the maritime Antarctic Peninsula exhibited broad tolerances towards light availability as well as euryhaline and eurythermal traits, far surpassing the environmental conditions of their respective habitats. In general, Antarctic organisms are expected to be rather stenohaline and stenotherm due to the long cold-water history of the Southern Ocean. However, maritime Antarctica is characterized by stronger seasonal and diurnal fluctuations in the abiotic parameters compared to continental Antarctica; hence, broader ecophysiological tolerances of the inhabiting biota might be assumed. Nevertheless, an important aspect that should be considered is related to the fact that all six benthic diatom species were grown as clonal cultures for >1 year under controlled lab conditions before the experiments were undertaken. It might be possible that the measured ecophysiological response patterns do not always reflect the in situ responses. In addition, due to the cultivation procedure described we can not rule out that we selected for the most tolerant species while sensitive taxa were outcompeted. Although logistically challenging in Antarctica, more field experiments are urgently needed to better understand the real world.

### 4.1. Light

Photosynthesis—the driving force for the energy metabolism and, hence, essential for the viability and survival of benthic diatoms—is primarily dependent on light availability. All six diatom species exhibited taxon-specific response patterns over a wide range of photon fluence rates, with only slight photoinhibition. Overall, the marine isolates exhibited higher NPP_max_ compared to the limnic ones. The highest photosynthetic rates were reached at low photon fluence rates, as reflected by low light compensation and light saturation points. All data clearly point to low light requirements for photosynthesis. In general, Antarctic diatoms are known for their fast growth in low-light conditions [65], because their photosynthesis seems to be especially shade-adapted [66]. The few data available on benthic diatoms from polar regions confirm a high photophysiological plasticity to acclimate to the prevailing, often very low light conditions [10,11,21,22,23,67]. In addition, this wide photophysiological plasticity seems to be a rather general trait of many diatom species [24], as documented in species from Arctic Kongsfjorden [27], but also in numerous species from the shallow waters of the temperate Baltic Sea [48,54].

Particularly for benthic diatoms, low light adaptation is crucial, since Antarctic microphytobenthic communities experience a strong seasonally changing light climate, often with low average photon fluence rates. During winter periods, with ice cover and short daylight periods, little or no light reaches the benthic diatoms—especially if the ice is covered by snow [68]. During summer, incident light penetration can be also reduced due to increased turbidity, which is driven by suspended particulate matter from glacial meltwater and riverine discharge [69]. In addition, wind- and organism-induced resuspension of the sediment can lead to a decline in the light availability through burial of the diatom cells. However, due to their motility, raphid diatoms are able to escape unfavorable low-light conditions in the sediment [70]. Vertical migration of benthic diatoms has been identified as an important behavioral trait to control the short-term variability of photosynthesis—at least in temperate regions. Although published studies on the vertical migration of benthic diatoms in Antarctica and the Arctic under polar day and night conditions are lacking, a few reports also indicate motility in polar species [27]. After the sea ice breakup in spring, solar radiation penetrates the coastal water column, with strong attenuation of the short wavelengths due to the prevailing optical properties, which are influenced by particle load from glaciers and yellow substances originating from meltwater and terrestrial runoff [69,71]. At 10 m depth in the inner Potter Cove, PAR was measured between 10 and 200 µmol photons m^−2^ s^−1^ in the winter and summer, respectively [28]. *Chamaepinnularia gerlachei* was sampled at 15 m depth in the inner Potter Cove, and showed a light compensation point of 15.3 μmol photons m^−2^ s^−1^ and a light saturation point of 59.8 μmol photons m^−2^ s^−1^, which fit well to the prevailing in situ light conditions.

An interesting aspect was the overall twofold-higher NPPmax exclusively in *Navicula criophiliforma* (about 200 µmol O_2_ mg^−1^ Chl *a* h^−1^) compared to all other studied Antarctic benthic diatom species. At present, we can only speculate to explain these data, but the largest cell size of *N. criophiliforma* (<52 × 8.5 µm, Table 2, Figure 2) among all species leads to the highest cell volume and, hence, to more chloroplasts and pigments. Recent data on the green microalga *Dunaliella teriolecta* experimentally prove that the established *package effect* theory, which predicts that larger phytoplankton cells should show poorer photosynthetic performance because of reduced intracellular self-shading, is challenged [72]. The latter authors reported that larger cells of *D. teriolecta* showed substantially higher rates of oxygen production along with higher chlorophyll values compared to smaller cells.

All six species could not only cope well with low-light conditions, but also showed high photosynthetic rates up to 1600 μmol photons m^−2^ s^−1^, with a minor-to-moderate degree of photoinhibition—especially in the marine strains. During the process of photoinhibition, diatoms are still able to perform photosynthesis without being completely inhibited. Excessive light is absorbed by the photosystems and harmlessly emitted via heat as a protective mechanism (non-photochemical quenching) for the photosynthetic apparatus [73]. Further exposure to excessive light, however, can lead to damage of the D1 protein, leading to a decrease in electron transfer [74]. All benthic diatom species exhibited low light requirements for photosynthesis combined with a pronounced photophysiological plasticity that also allowed broad tolerance to high-incident-light conditions.

### 4.2. Temperature

Photosynthesis, respiration, and growth, along with their underlying enzymatic mechanisms, are strongly controlled by temperature. Therefore, reductions in photosynthetic and respirational activity, as well as in growth under saturated light conditions, are a consequence of inhibition of the most temperature-sensitive enzymes. Low temperatures slow down electron transport, thereby decreasing the ability to use photons for photochemically produced energy. High temperatures can influence the photorespiration activity of RuBisCO (ribulose-1,5-bisphosphat-carboxylase/-oxygenase) by removing its specificity towards CO_2_ binding rather than that of O_2_, thereby increasing energy demand [75]. Similar to other studies using the same methodological approach on Baltic Sea benthic diatoms [48,54], the photosynthetic and respirational rates of Antarctic diatom strains seemed to also be decoupled from one another, with respiration always showing optima at higher temperatures compared to photosynthesis. For temperate diatoms, but also for terrestrial green algae, temperature requirements for respiration and photosynthesis differ, as explained by the higher dependency of photosynthesis on light, while temperature-dependent enzymatic activities mainly control respiration [76,77]. A more recent study partially confirmed that light-dependent photosynthetic reactions are indeed unaffected by temperature, while the carbon fixation reactions are driven by temperature [78]. Furthermore, respiratory and photosynthetic activities in diatoms are strongly coupled, which is mechanistically explained by tight physical interactions between mitochondria and chloroplasts [79]. Consequently, light stimulates respiration, resulting in an optimal ATP/NADPH ratio for subsequent carbon dioxide fixation by RuBisCO.

Another important aspect is the observation that the optimal temperature for photosynthesis (Figure 9, 20 °C) was higher compared to that for growth (Figure 10, <15 °C). These differences in both physiological processes can be explained by the exposure time to the stressor temperature. The time scale of stress is relevant, as algae may cope temporarily with strong temperature stress if acting only for hours to days, and may subsequently recover from damage under optimal conditions [80]. However, on a longer time scale (weeks), the algae experience progressively more impaired cellular processes until the upper temperature for survival is reached. Consequently, temperature optima for photosynthesis are often higher than those for growth, because both physiological processes are not directly coupled and, hence, photosynthesis does not necessarily match the temperature–growth pattern. In addition, growth is a more general physiological process that integrates all positive and negative influences of temperature on the whole metabolism [81]. The data shown clearly indicate broad temperature tolerance of photosynthesis and respiration in the Antarctic benthic diatom isolates, far exceeding in situ temperatures in their respective habitats. While the temperature tolerance of Antarctic phytoplankton—which usually do not survive temperatures > 8–9 °C, and which is consistent with the maximum in situ temperature—is recognized as stenotherm [82,83], benthic diatoms in shallow waters or in tidal pools during the polar day can be exposed to temperatures that are several times higher compared to the water column. For Potter Cove, where the investigated strains were sampled, the tides are semi-diurnal, and the temperature in some tidal pools may change from 2 to 14 °C within only 8 h [37].

As already mentioned in the introduction, all benthic diatoms from the Arctic that have so far been experimentally studied under controlled conditions typically exhibit eurythermal and psychrotolerant traits, while those from Antarctica show stenothermal and psychrophilic features [27,28]. These fundamental differences in the response patterns can be explained by the geologically 10-fold longer cold-water history of Antarctica compared to the Arctic, fostering adaptive and evolutionary processes in the inhabiting organisms, which finally led to many endemic marine organisms in Antarctica [32]. However, in sharp contrast to the data of Longhi et al. [28], all six benthic diatom species in the present study exhibited very similar ecophysiological response patterns, comparable not only to those from their Arctic counterparts, but also to those from temperate regions such as the Baltic Sea [48,54], hence pointing to eurythermal and psychrotolerant traits. The unexpectedly broad temperature tolerances are not easy to explain, but Potter Cove is one of the few places in Antarctica where long-term ecological observational data exist. Based on >20-year time series of sea surface temperature, data prove a temperature increase of 0.7 to 0.8 °C in the last two decades, accelerating biological activities and physicochemical processes in the shallow coastal waters of Potter Cove [84]. As a consequence, summer meltwater runoff from coastal ice sheets and from thawing of coastal permafrost areas causes freshening of the shallow water, along with increasing turbidity due to mobilization of lithogenic particles, so that benthic biota are strongly affected by such highly dynamic and new climate-sensitive environmental conditions [84]. Therefore, it is reasonable to assume that during the ca. 20-year time span between the study of Longhi et al. [28] and the data presented here, changes within the benthic diatom community took place, i.e., from more stenoecious (endemic) to euryoecious (non-endemic) taxa. It might also be possible that non-endemic benthic diatoms invaded the Antarctic Peninsula from sub-Antarctic islands and from South America—for example, as hull biofouling organisms—as shown for other benthic organisms, such as invertebrates [85]. However, comprehensive information on the biodiversity and biogeography of marine benthic diatoms in Antarctica is still lacking, while freshwater species are very well studied [34].

### 4.3. Salinity

In general, the photosynthetic potential of the six benthic diatom species exhibited broad tolerances, with habitat-typical salinity optima of 25 to 35 S_A_ for the marine strains and 1 to 10 S_A_ for the freshwater strains. Due to the topographic division within Potter Cove and the freshwater runoff, salinity in the inner part of the bay exhibits lower salinities > 29.6 S_A_, compared to the outer part, which has fully marine salinities [38]. Salinity stress is related to the toxic effects of Na^+^ and Cl^−^, and often results in a decline in photosynthetic activity or a change in PS II efficiency [86,87,88,89]. This can lead to oxidative stress, consequently damaging lipid membranes, proteins, or nucleic acids [90], while also interfering with the photosynthetic and respiratory electron transport. The accompanying effect of changing cell volumes can also lead to the deactivation of the photosynthetic apparatus [88].

In the marine rock pools, benthic diatoms are typically exposed to strong tidal-induced salinity changes, as they are cut off from the main body of marine water. In the rock pools, salinity can increase as a result of strong evaporation due to insolation and wind, or decrease because of precipitation or glacial freshwater inflow [91]. The marine strain *N. criophiliforma*, sampled from a rock pool in Potter Cove, exhibited a wide euryhaline tolerance, with >20% photosynthetic potential between 10 and 55 S_A_, thereby able to cope well with this abiotic factor. The remaining two marine species were also euryhaline in terms of photosynthesis. Such broad tolerance ranges are typically found in diatoms living under and within the sea ice, and which can cope with salinities of up to 60–100 S_A_ [87]. However, apart from sea-ice diatoms, there exist only fragmentary data on salinity responses in polar benthic diatoms. For the Arctic *Nitzschia* cf. *aurariae*, growth between 15 and 45 S_A_, with an optimum at 20 to 40 S_A_, was reported, and it was thus characterized as moderately euryhaline [92]. In contrast to polar benthic diatoms, their temperate counterparts are well studied in terms of a commonly wide salinity tolerance. Numerous benthic diatoms from the North Sea exhibited high growth rates between 2 and 45 S_A_ [93], and between 10 and 40 S_A_ [94], while a study from the Baltic Sea reported growth between 1 and 50 S_A_ [95].

The underlying mechanisms of osmotic acclimation have not yet been studied in Antarctic benthic diatoms. In contrast, ice-associated diatoms trapped in the brine channels can experience salinities three times that of seawater. These algae typically synthesize and accumulate high concentrations of organic osmolytes and compatible solutes in response to hypersaline stress, such as proline, mannitol, glycine betaine, and/or dimethylsulfoniopropionate (DMSP) [96].

## 5. Conclusions

In conclusion, all six benthic diatom species isolated from the Antarctic Peninsula exhibited strong euryhaline and eurythermal traits far surpassing the environmental conditions of their respective habitats. Pronounced low light requirements and species-specific photophysiological plasticity with minor photoinhibition were present. With regard to the ongoing climate change—particularly in maritime Antarctica—the increasing water temperatures of Potter Cove, and the accompanying fluctuations in salinity and the light field, all of the isolates seemed to be well acclimated, as reflected in their eurythermal and euryhaline response patterns.

## Figures and Tables

**Figure 1 genes-13-01264-f001:**
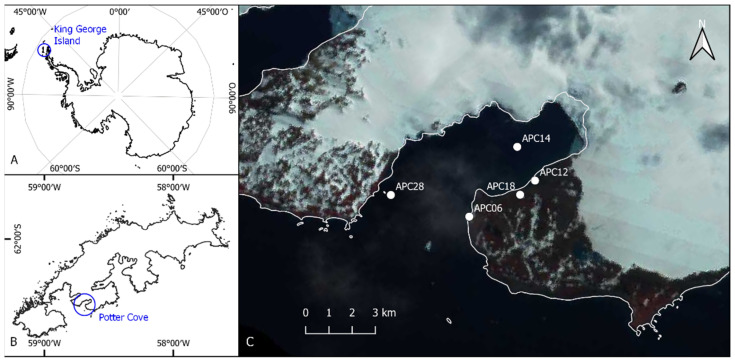
Sampling points: (**A**) map of the Antarctic, (**B**) map of King George Island, and (**C**) sample points in the Potter Cove: limnic location APC18, brackish water location APC12, and marine locations APC06, APC14, and APC28. Basemap: Landsat image mosaic of Antarctica (LIMA).

**Figure 2 genes-13-01264-f002:**
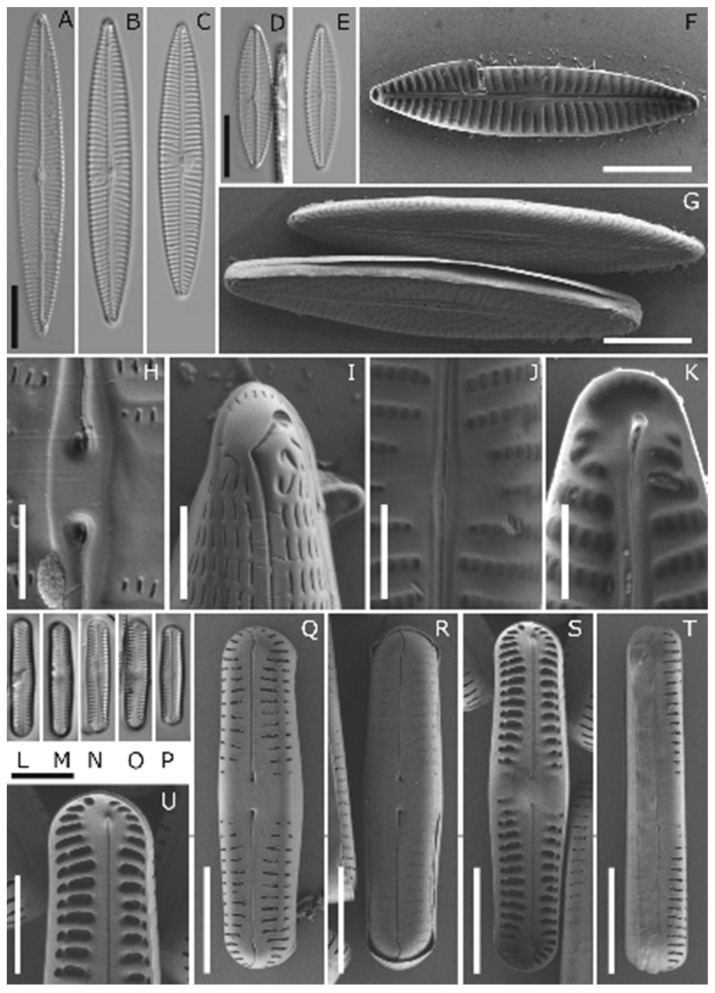
LM and SEM Image Set 1 (**A**–**U**): (**A**–**K**) *Navicula criophiliforma*. (**A**–**E**) LM pictures; development of auxospores led to size differences in the valves in the culture. (**F**–**K**) SEM pictures: (**F**) whole-valve internal view, (**G**) whole-valve external view, (**H**) external central raphe endings, (**I**) valve apex external view, (**J**) internal proximal raphe endings, (**K**) valve apex internal view. (**L**–**T**) *Chamaepinnularia gerlachei*, APC12 D294_006, and *Chamaepinnularia gerlachei*: (**L**,**M**) LM pictures of the strain *Chamaepinnularia gerlachei*, (**N**–**P**) LM pictures of the strain APC12 D294_006, (**Q**–**U**) SEM pictures of APC12 D294_006. (**Q**,**R**) whole-valve external view, hymenate occlusion of areolae partly corroded, (**S**) whole-valve internal view, (**T**) valve in girdle view, (**U**) valve apex internal view. Scale bars: (**A**–**G**) and (**L**–**P**) 10 µm, (**H**–**K**) and (**U**) 2 µm, and (**Q**–**T**) 5 µm.

**Figure 3 genes-13-01264-f003:**
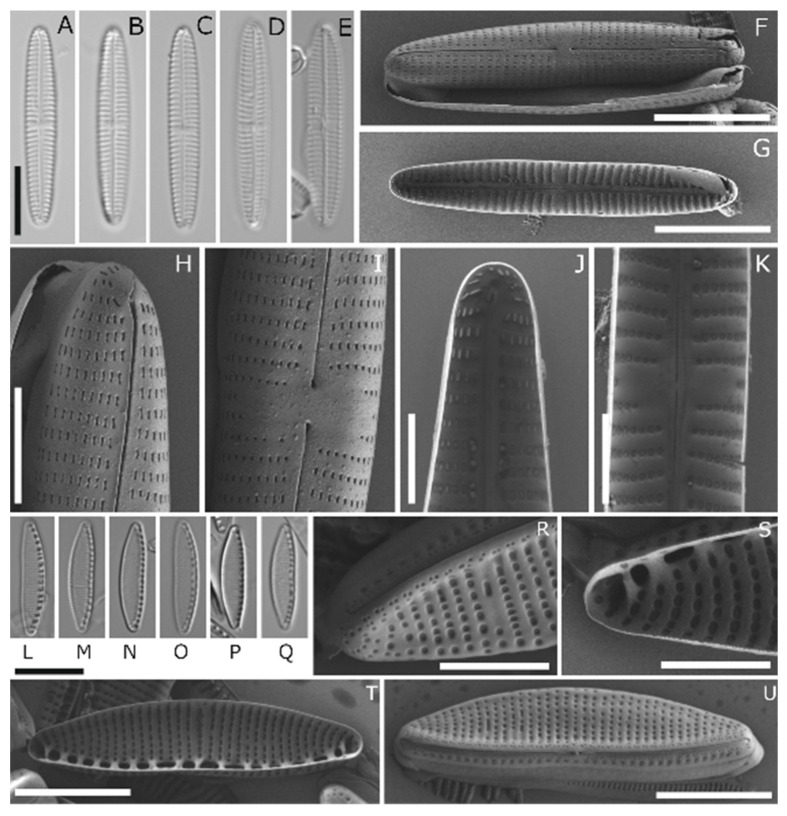
LM and SEM Image Set 2 (**A**–**U**): (**A**–**K**) *Navicula concordia*. (**A**–**E**) LM pictures. (**F**–**K**) SEM pictures: (**F**) whole-valve external view, (**G**) whole-valve internal view, (**H**) valve apex external view, (**I**) external proximal raphe endings, (**J**) valve apex internal view, (**K**) internal proximal raphe endings. (**L**–**U**) *Nitzschia annewillemsiana*: (**L**–**Q**) LM pictures. (**R**–**U**) SEM pictures: (**R**) valve apex external view, (**S**) valve apex internal view, (**T**) whole-valve internal view, (**U**) whole-valve external view. Scale bars: (**A**–**G**) and (**L**–**Q**) 10 µm, (**H**–**K**) and (**R**,**S**) 3 µm, and (**T**,**U**) 5 µm.

**Figure 4 genes-13-01264-f004:**
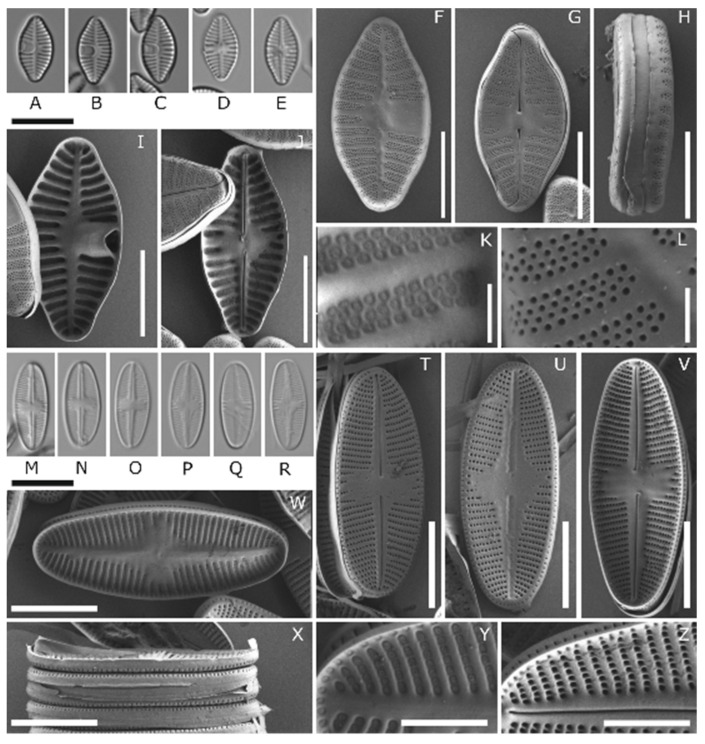
LM and SEM Image Set 3: (**A**–**L**) *Planothidium* sp. (**A**–**E**) LM pictures. (**F**–**L**) SEM pictures: (**F**) whole-sternum-valve external view, (**G**) whole-raphe-valve external view, (**H**) valve in girdle view, (**I**) whole-sternum-valve internal view, (**J**) whole-raphe-valve internal view, (**K**) internal valve view of one stria with rows of areolae with hymenate occlusions, (**L**) internal valve view of one stria with rows of areolae. (**M**–**Z**) *Psammothidium papilio*: (**M**–**R**) LM pictures. (**T**–**Z**) SEM pictures: (**T**) whole-raphe-valve external view, (**U**) whole-sternum-valve external view, (**V**) whole-raphe-valve-internal view, (**W**) whole-sternum-valve internal view, (**X**) valves in girdle view, (**Y**) internal sternum valve view of areolae with hymenate occlusions, (**Z**) internal raphe valve view of areolae. Scale bars: (**A**–**E**), (**F**–**J**) and (**T**–**X**) 5 µm, (**K**,**L**) 1 µm, and (**Y**,**Z**) 2 µm.

**Figure 5 genes-13-01264-f005:**
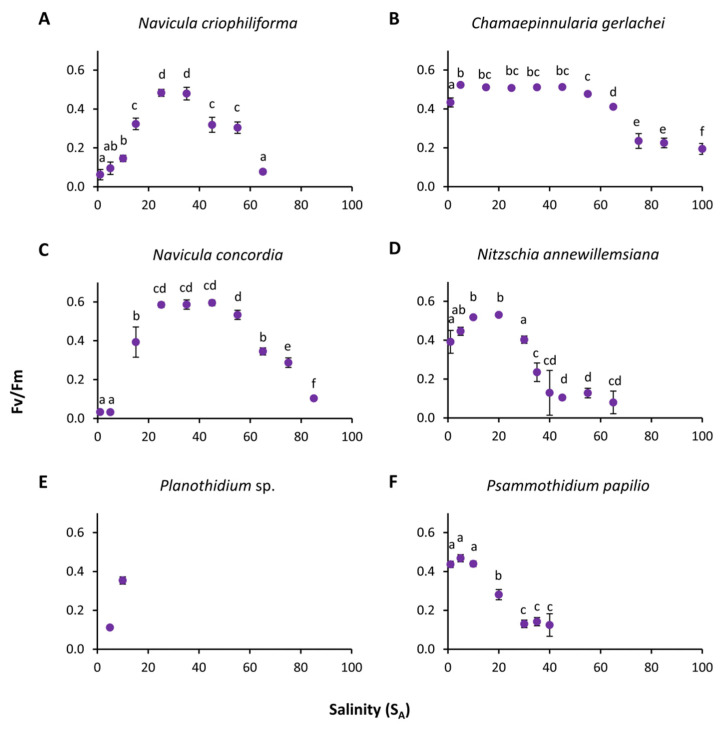
Effective quantum yield of photosystem II (Fv/Fm) as a function of salinity of six benthic diatom strains from Antarctica after 3 days of incubation (**A**–**F**). Data represent mean values ± SD (*n* = 6). Different lowercase letters indicate significant means (*p* < 0.05; one-way ANOVA with post hoc Tukey’s test). (**A**) *Navicula criophiliforma*, (**B**) *Chamaepinnularia gerlachei*, (**C**) *Navicula concordia*, (**D**) *Nitzschia annewillemsiana*, (**E**) *Planothidium* sp., and (**F**) *Psammothidium papilio*.

**Figure 6 genes-13-01264-f006:**
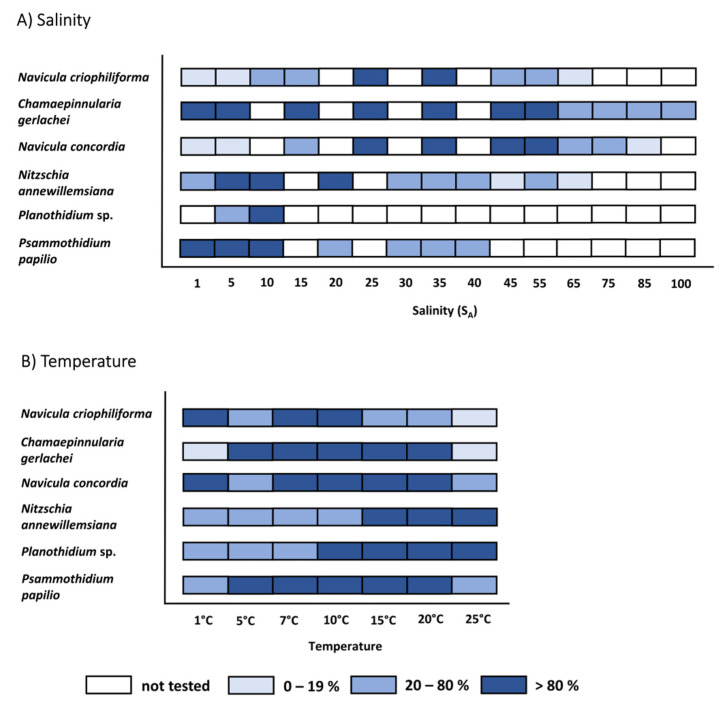
Effects of (**A**) salinity and (**B**) temperature on the effective quantum yield of photosystem II (Fv/Fm) of six benthic diatom strains from Antarctica. Dark blue symbols represent the range of the highest effective quantum yield at the 80th percentile and above, medium blue symbols are between the 20th and 80th percentiles, light blue symbols represent the 20th percentile and below, and white symbols were not tested. Data represent mean values (*n* = 6).

**Figure 7 genes-13-01264-f007:**
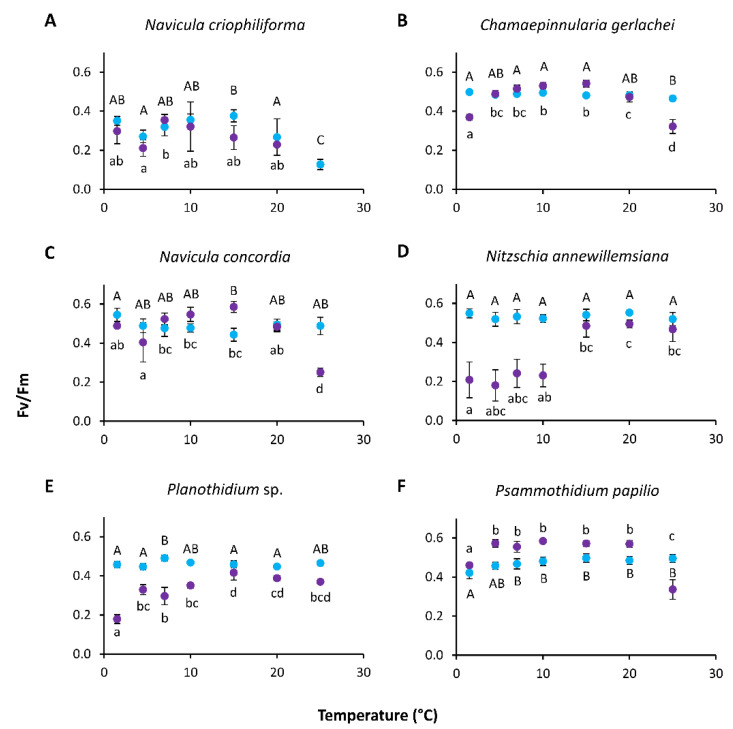
Effective quantum yield of photosystem II (Fv/Fm) as a function of temperature of six benthic diatom strains from Antarctica after 0 days (blue) and 5 days (purple) of incubation (**A**–**F**). Data represent mean values ± SD (*n* = 6). Different capital (t_0_) and lowercase (t_5_) letters indicate significant means (*p* < 0.05; one-way ANOVA with post hoc Tukey’s test). (**A**) *Navicula criophiliforma*, (**B**) *Chamaepinnularia gerlachei*, (**C**) *Navicula concordia*, (**D**) *Nitzschia annewillemsiana*, (**E**) *Planothidium* sp., and (**F**) *Psammothidium papilio*.

**Figure 8 genes-13-01264-f008:**
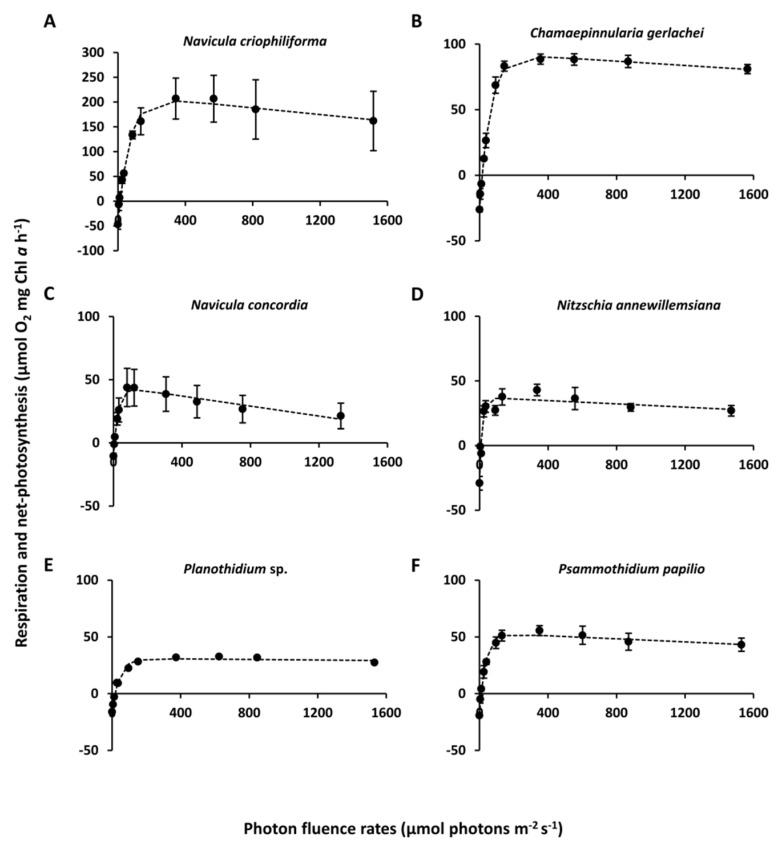
Photosynthesis and respiration rates (μmol O_2_ mg^−1^ Chl *a* h^−1^) as a function of increasing photon flux density (μmol photons m^−2^ s^−1^) of six benthic diatom strains from Antarctica kept at 8 °C in f/2 medium: 33 S_A_ (**A**–**C**) and 1 S_A_ (**D**–**F**). Data represent mean values ± SD (*n* = 3). Data points were fitted using the model of Walsby [50]. (**A**) *Navicula criophiliforma*, (**B**) *Chamaepinnularia gerlachei*, (**C**) *Navicula concordia*, (**D**) *Nitzschia annewillemsiana*, (**E**) *Planothidium* sp., and (**F**) *Psammothidium papilio*.

**Figure 9 genes-13-01264-f009:**
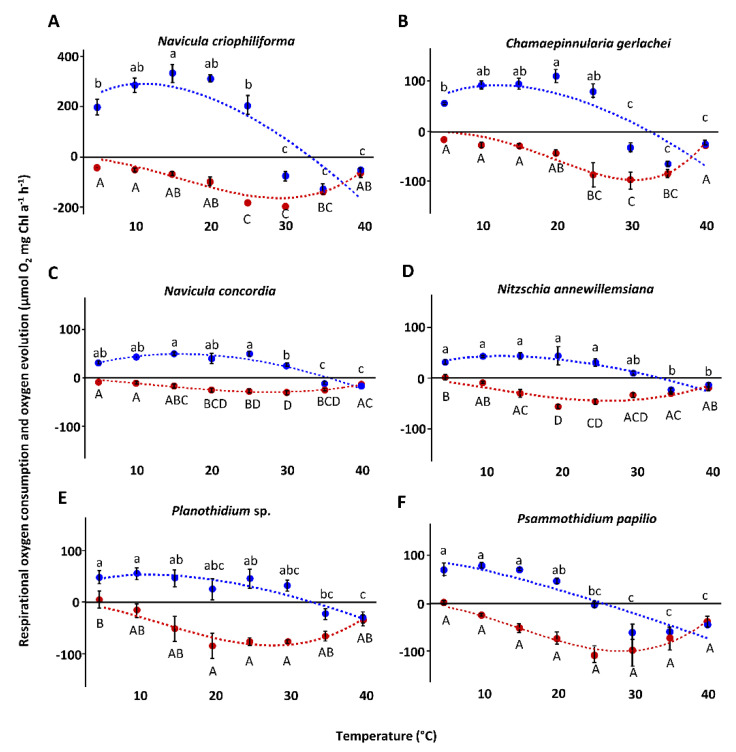
Photosynthetic (blue) oxygen production at 342 ± 40 µmol photons m^−2^ s^−1^ and respiratory (red) oxygen consumption in darkness of six benthic diatom strains from Antarctica, as a function of increasing temperature (**A**–**F**). The measured data were fitted using the model of Yan and Hunt [55] (photosynthesis: blue dashed line; respiration: red dashed line). All cultures were kept in f/2 Baltic Sea media: 33 S_A_ (**A**–**C**) and 1 S_A_ (**D**–**F**). Data represent mean values ± SD (*n* = 3). Different lowercase (photosynthesis) and capital letters (respiration) indicate significant means (*p* < 0.05; one-way ANOVA with post hoc Tukey’s test). (**A**) *Navicula criophiliforma*, (**B**) *Chamaepinnularia gerlachei*, (**C**) *Navicula concordia*, (**D**) *Nitzschia annewillemsiana*, (**E**) *Planothidium* sp., and (**F**) *Psammothidium papilio*.

**Figure 10 genes-13-01264-f010:**
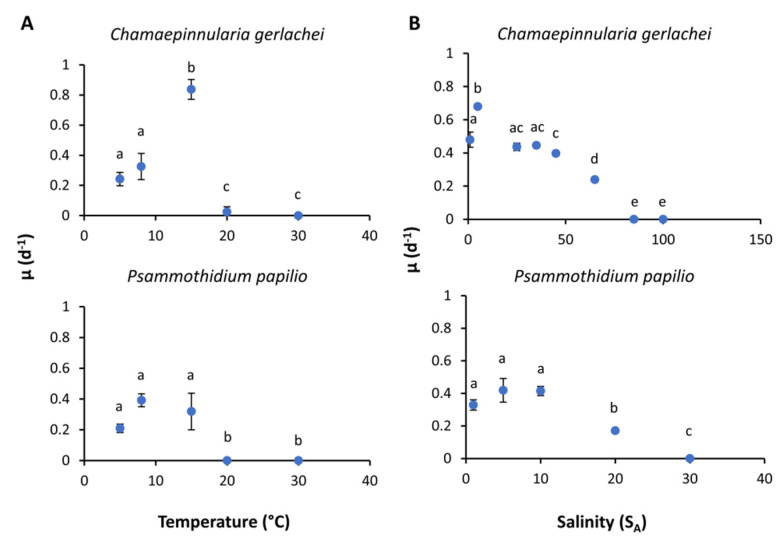
Growth rates (µ d^−1^) in relation to (**A**) temperature and (**B**) salinity of the respective diatom strains *Chamaepinnularia gerlachei* and *Psammothidium papilio*. Data represent mean values ± SD (*n* = 3). Different lowercase letters represent significance levels among all means, as calculated per temperature or salinity by one-way ANOVA (Tukey’s test, *p* < 0.05). Please note the different salinity ranges for both species.

**Table 1 genes-13-01264-t001:** List of sample locations at Carlini Station, King George Island, Potter Cove, in austral summer 2020 (January/February), with information on sample site, altitude or water depth, sample date, collector georeferenced, and sampled substrate; a.s.l.: above sea level.

Sample Location	Site	Sample Origin	Altitude/ Water Depth	Date of Sampling	Collector	Georeference
APC06	Potter Cove, coast at Penon 0	Marine	0 m	29 January 2020	J. Zimmermann	S 62°14′30.55″, W 58°40′54.96″
APC12	Potter Cove, coast east of Carlini Station	Brackish	0 m	30 January 2020	J. Zimmermann	S 62°14′07.78″, W 58°39′27.91″
APC14	Potter Cove, Island A4	Marine	15 m depth	31 January 2020	J. Zimmermann, G.L. Campana, Diver Team	S 62°13′43.61″, W 58°39′49.36″
APC18	Potter Cove, drinking water reservoir	Freshwater	51 m a.s.l.	1 February 2020	J. Zimmermann	S 62°14′16.30″, W 58°39′44.10″
APC28	Potter Cove, coast at Penon de Pesca	Marine	5 m depth	7 February 2020	J. Zimmermann, G.L. Campana, Diver Team	S 62°14′16.5″, W 58°42′44.2”

**Table 2 genes-13-01264-t002:** List of strains established from Antarctic marine and freshwater samples collected at Carlini Station, King George Island, Potter Cove, in austral summer 2020 (January/February), with scientific name, information on dimensions of the valves, striae density, and sequenced marker genes. RV: raphe valve, SV: sternum valve.

Strain	Scientific Name	Marine/ Freshwater	Length (µm)	Width (µm)	Striae in 10 µm	Marker Genes
APC14 D296_001	*Chamaepinnularia gerlachei*	Marine	17.1–20.6	4.1–5.4	18–20	whole 18 S, *rbc*L
APC06 D288_003	*Navicula* *criophiliforma*	Marine	24.2–52.4	5.8–8.5	11–12	18 SV4, *rbc*L
APC28 D310_004	*Navicula* *concordia*	Marine	29.5–30.5	4.7–5.3	13–14	18 SV4, *rbc*L
APC18 D300_012	*Nitzschia annewillemsiana*	Freshwater	15.2–17.1	3.6–4.1	25–26	18 SV4, *rbc*L
APC18 D300_015	*Planothidium* sp.	Freshwater	10.9–11.3	5.6–6.1	16–18 (RV) 17–18 (SV)	18 SV4, *rbc*L
APC18 D300_023	*Psammothidium papilio*	Freshwater	13.8–14.7	5.4–5.9	28–30 (RV) 26–30 (SV)	18 SV4, *rbc*L

**Table 3 genes-13-01264-t003:** Parameters of respective P–I curves (Figure 8) of six benthic diatom species (*n* = 3) kept at 8 °C. Different lowercase letters represent significance levels among all means as calculated by one-way ANOVA (Tukey’s test, *p* < 0.05). NPP_max_ represents the maximal oxygen production rate, α is the initial slope of production in the light-limited range, β is the terminal slope of production in extensive light range (photoinhibition), I_k_ is the light saturation point, and I_c_ is the light compensation point.

Species	NPP_max_ (µmol O_2_ mg^−1^ Chl *a* h^−1^)	Respiration (µmol O_2_ mg^−1^ Chl *a* h^−1^)	α (µmol O_2_ mg^−1^ Chl *a* h^−1^) (µmol Photons m^−2^ s^−1^)^−1^	β (µmol O_2_ mg^−1^ Chl *a* h^−1^) (µmol Photons m^−2^ s^−1^)^−1^	I_k_ (µmol Photons m^−2^ s^−1^)	I_c_ (µmol Photons m^−2^ s^−1^)	NPP_max_: Respiration
*Navicula* *criophiliforma*	202.3 ± 45.4 a	−47 ± 8.9 a	3.9 ± 0.4 a	−0.03 ± 0.02 a	64 ± 11.5 a	13.4 ± 1.4 ab	4.3 ± 0.9 a
*Chamaepinnularia* *gerlachei*	90.3 ± 4.1 b	−26.2 ± 0.8 b	2 ± 0.1 b	−0.01 ± 0.00 bc	59.8 ± 1.7 a	15.3 ± 0.2 a	3.5 ± 0.1 ab
*Navicula* *concordia*	42 ± 14.5 bc	−10.5 ± 3.1 c	2 ± 0.6 b	−0.02 ± 0.01 ac	25.9 ± 2.7 bc	5.8 ± 1 c	4 ± 0.4 a
*Nitzschia annewillemsiana*	36.6 ± 5.4 bc	−25.9 ± 3.7 b	3.8 ± 0.69 a	−0.01 ± 0.00 bc	16.3 ± 3.9 b	8.7 ± 0.3 cd	1.4 ± 0 c
*Planothidium* sp.	30.7 ± 0.5 c	−16.0 ± 6.2 bc	1.1 ± 0.3 b	0.0 ± 0.0 b	23.6 ± 8.7 bc	17.5 ± 3 a	1.9 ± 0.4 c
*Psammothidium* *papilio*	52.2 ± 5 bc	−19.2 ± 1.7 bc	2.1 ± 0.2 b	−0.00 ± 0.00 bc	33.7 ± 1.8 c	10.6 ± 0.2 bd	2.7 ± 0.1 b

**Table 4 genes-13-01264-t004:** Results of model calculation for temperature-dependent growth rate, photosynthetic rate, respirational rate, and salinity-dependent growth rate, following the model of Yan and Hunt [55].

			*Navicula criophiliforma*	*Chamaepinnularia gerlachei*	*Navicula concordia*	*Nitzschia annewillemsiana*	*Planothidium* sp.	*Psammothidium papilio*
**Growth** **(salinity)**	Maximal growth rate		-	0.58	-	-	-	0.42
	Optimal salinity		-	6.53	-	-	-	5.28
	Maximal salinity		-	93.69	-	-	-	29.26
	Residual sum of squares		-	0.0884	-	-	-	0.03172
	Salinity range for	Optimal growth (80% growth rate)	-	0.13–31.79	-	-	-	0.90–13.71
		Growth (20% growth rate)	-	0.00–79.23	-	-	-	0.00–25
**Growth** **(temperature)**	Maximal growth rate		-	0.44	-	-	-	0.30
	Optimal temperature		-	12.96	-	-	-	6.48
	Maximal temperature		-	28.85	-	-	-	6.53
	Residual sum of squares		-	0.9345	-	-	-	5.28
	Temperature range for	Optimal growth (80% growth rate)	-	6.48–19.89	-	-	-	1.56–14.47
		Growth (20% growth rate)	-	0.90–27.11	-	-	-	0.00–25.10
**Photosynthesis**	Maximal photosynthetic rate		292.59	91.37	48.93	43.46	53.49	85.82
	Optimal temperature		11.12	12.08	15.66	12.48	11.52	2.99
	Maximal temperature		33.35	32.47	35.63	33.49	30.30	26.03
	Residual sum of squares		199,464	31,639	2012	4575	13,940	10,575
	Temperature range for	Optimal photosynthesis (80% photosynthetic rate)	4.11–20.14	5.01–20.56	7.67–24.32	5.19–21.23	4.44–20.43	0.2–10.05
		Photosynthesis (20% photosynthetic rate)	0.2–30.68	0.37–30.1	0.99–33.34	0.38–31.04	0.25–30.71	0–22.48
**Respiration**	Maximal respirational rate		−185.99	−97.77	−28.98	−45.46	−84.06	−100.19
	Optimal temperature		29.67	30.59	27.65	26.61	28.04	28.66
	Maximal temperature		41.9	41.17	44.44	42.88	42.97	42.49
	Residual sum of squares		24,352	8086	343.9	2815	13,143	8103
	Temperature range for	Optimal respiration (80% respirational rate)	22.26–35.62	23.91–35.79	18.57–35.57	17.83–34.28	19.62–35.16	20.64–35.31
		Respiration (20% respirational rate)	10.34–40.77	12.4–40.21	6.33–42.8	6.03–41.3	7.51–41.54	8.55–41.18

## Data Availability

Vouchers and DNA of all strains were deposited in the collections at Botanischer Garten und Botanisches Museum Berlin, Freie Universität Berlin (B). DNA samples were stored in the Berlin DNA bank, and are available via the Genome Biodiversity Network (GGBN). All sequences were submitted to the European Nucleotide Archive (ENA, http://www.ebi.ac.uk/ena/). All cultures are available from the authors at the culture collection of the Department of Applied Ecology and Phycology, University of Rostock.

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
