# Peer review of "Photosynthetic, Respirational, and Growth Responses of Six Benthic Diatoms from the Antarctic Peninsula as Functions of Salinity and Temperature Variations"

_genes, 2022, doi:10.3390/genes13071264_

Round 1
Reviewer 1 Report
The manuscript submitted by Prelle and collaborators describes several physiological aspects, including photosynthesis, respiration and growth, of six benthic diatoms species from the Antarctic Peninsula. The authors start the article describing the zones were and all the six diatoms were found and isolated and the morphological characteristics of the species. Then they study the effects of varying temperature and salinity on photosynthesis, respiration and cell growth.
This article provides a substantial contribution to the field by exploring the adaptive responses of diatom species isolated from specific sampling points, including marine and freshwater stations, in an Antarctic region that has been severely affected by global warming over the past 50 years. In addition, this work puts also in evidence the strong resilience of diatoms form the Antarctic to both temperature and salinity, in contrast to what was previously shown for other phytoplankton.
The article is well written and the methodology is fairly correct and well explained. However, I highlight below some aspects that I suggest to be discussed:
- The discussion can be improved by adding more comparisons with other diatom species from other latitudes. How different is the data reported here to other works using diatoms from more template latitudes or with Arctic species?
- I was surprise to see that the diatom N. criophiliforma has a much higher photosynthetic yields compared to the other species studied, however, the authors overlooked this on the discussion. Do the authors have any idea to explain why this difference and whether it can mean an advantageous adaptive trait of this diatom species? Is it related to the location where this species was found? Did the authors observe a difference on the abundance of this species over others?
- It is not clear why the authors chose to use the only two “exemplary” diatoms to perform their growth analyzes; is there another criteria apart of being one limnic and one marine? In addition, they show that in both species, cell growth is completely inhibited at >20C, however in all species there is no significant effect on photosynthesis yield, oxygen evolution and respiration, at 20C compared to lower temperatures. This could mean that other factors are much more determinant on their sensitivity to such temperatures; could the authors discuss on what can be those factors?
Minor:
- In order to facilitate the interpretation of the results, I recommend the authors to include a zooming of the areas of lower light intensities and respiration.
- In the description of figure 9 “umol photons” is missing as light intensity.
Author Response
Dear editor and reviewer 1,
many thanks for your encouraging and constructive comments on our manuscript. We deeply appreciate that you all like the scientific story. We tried our best to address your questions and comments. Below please find our point-to-point responses.
Referee 1:
The discussion can be improved by adding more comparisons with other diatom species from other latitudes. How different is the data reported here to other works using diatoms from more template latitudes or with Arctic species?
We added to the discussion on light:
In addition, this wide photo-physiological plasticity seems to be a rather general trait of many diatom species [24], as documented in species from Arctic Kongsfjorden [27], but also in numerous species from the shallow waters of the temperate Baltic Sea [48,54].
The discussion on temperature is mainly fine from our side, but we highlighted the following sentence:
However, in sharp contrast to the data of Longhi et al. [28], all six benthic diatom species in the present study exhibited very similar ecophysiological response patterns comparable not only to those from their Arctic pendants, but also to those from temperate regions such as the Baltic Sea [48,54], and hence point to eurythermal and psychrotolerant traits.
In contrast to light and temperature, salinity effects on benthic diatoms in polar waters are still poorly studied, while there exist plenty of data from temperate regions. We added these sentences:
In contrast to polar benthic diatoms, their temperate pendants are well studied in terms of a commonly wide salinity tolerance. Numerous benthic diatoms from the North Sea exhibited high growth rates between 2 to 45 SA [93] and between 10 to 40 SA [94], while a study from the Baltic Sea reported growth between 1 to 50 SA [95].
I was surprise to see that the diatom N. criophiliforma has a much higher photosynthetic yields compared to the other species studied, however, the authors overlooked this on the discussion. Do the authors have any idea to explain why this difference and whether it can mean an advantageous adaptive trait of this diatom species? Is it related to the location where this species was found? Did the authors observe a difference on the abundance of this species over others?
Good point – here is our interpretation:
An interesting aspect was the overall two-fold higher NPPmax exclusively in Navicula criophiliforma (about 200 µmol O2 mg-1 Chl a h-1) compared to all other studied Antarctic benthic diatom species. At present, we can only speculate to explain these data, but the largest cell size of N. criophiliforma (< 52 x 8.5 µm, Table 2, Figure 2) among all species, will lead to the highest cell volume and hence to more chloroplasts and pigments. Recent data on the green microalga Dunaliella teriolecta experimentally prove, that the established package effect theory, which predicts that larger phytoplankton cells should show poorer photosynthetic performance because of reduced intracellular self-shading, is challenged [72]. The latter authors reported that larger cells of D. teriolecta showed substantially higher rates of oxygen production along with higher chlorophyll values compared to smaller cells.
It is not clear why the authors chose to use the only two “exemplary” diatoms to perform their growth analyzes; is there another criteria apart of being one limnic and one marine? In addition, they show that in both species, cell growth is completely inhibited at >20C, however in all species there is no significant effect on photosynthesis yield, oxygen evolution and respiration, at 20C compared to lower temperatures. This could mean that other factors are much more determinant on their sensitivity to such temperatures; could the authors discuss on what can be those factors?
Due to time constraints (BSc and MSc thesis work) only 2 species could be carefully tested for growth. Growth measurements require much more time and preparation in terms of methodology (weeks to months) compared to photosynthesis (days). But the referee is completed right that growth and photosynthesis responses under increasing temperatures strongly differed, which might be explained by the factor time (duration of exposure). We added the following to the discussion:
Another important aspect is the observation that the optimum temperature for photosynthesis (Figure 9, c. 20°C) was higher compared to that for growth (Figure 10, < 15°C). These differences in both physiological processes can be explained by the exposure time to the stressor temperature. The time-scale of stress is relevant as algae may cope temporarily with strong temperature stress if acting only for hours to days and subsequently may recover from damage at optimal conditions [80]. However, on a longer time-scale (weeks) the algae experience progressively more impaired cellular processes until the upper temperature for survival is reached. Consequently, temperature optima for photosynthesis are often higher than those for growth because both physiological processes are not directly coupled, and hence photosynthesis does not necessarily match the temperature-growth pattern. In addition, growth is a more general physiological process that integrates all positive and negative influences of temperature on the whole metabolism [81].
Minor:
In order to facilitate the interpretation of the results, I recommend the authors to include a zooming of the areas of lower light intensities and respiration.
Concerning this comment we feel that the paper is already quite long, and that the combination of data presented in Figure 8 and Table 3 is sufficient to fully reveal all necessary results on the PI curves.
In the description of figure 9 “umol photons” is missing as light intensity.
We corrected the missing “µmol photons”
We hope that the reviewer is satisfied with our revision, and now agrees on acceptance.
Best wishes
Ulf Karsten & co-authors
Reviewer 2 Report
In this manuscript, Prelle et. al., explored the six diatom strains isolated from the Antarctic Peninsula, and investigated their Photosynthetic, respirational and growth patterns under varying light availability, temperature, and salinity, which suggest the importance of ecophysiological plasticity in those selected species under the global warming context, especially in the fragile Antarctica.
After reading the whole article, I was impressed by the detailed methods, logical discussion and sound results. Actually, the authors have contributed some relevant research in this area, for example another recent one in MDPI Microorganisms https://doi.org/10.3390/microorganisms10040749, I appreciate the efforts, especially for the experiments designed for the ecophysiology analysis, that authors have contributed and believe it is useful and interesting for the polar community to read.
However, I do have some concerns with regarding to the completeness of background references for broad readers.
L62-64: “while similar studies for Antarctica are missing. ” “As for most other groups of eukaryotic microorganisms remarkably little is still known about marine benthic diatom biodiversity and ecophysiology in Antarctica. “
I think this paragraph shall be extended with more relevant and recent references.
- I was surprised the model polar diatom Fragilariopsis cylindrus (Thomas Mock et.al., Nature, 2017 https://www.nature.com/articles/nature20803) was not mentioned in the article. Please also refer to the latest review paper Diatoms and Their Microbiomes in Complex and Changing Polar Oceans, https://doi.org/10.3389/fmicb.2022.786764.
- Also, there are many emerging eukaryotic polar model species in recent years, for example the Chlamydomonas nivalis (Brown et al. 2015 https://www.tandfonline.com/doi/full/10.1657/AAAR0014-071 ), Chlamydomonas sp. ICE-L (Zhang et.al., 2020, DOI: 10.1016/j.cub.2020.06.029); Chlamydomonas sp. UWO241 (Zhang et.al., 2021 https://doi.org/10.1016/j.isci.2021.102084)
- Some recent reviews have also provided in-depth synthesis of research on photophysiology of polar phototrophic organisms to date, which is for authors’ considerations, such as Hüner et al. (2022) https://doi.org/10.1016/j.jplph.2021.153557 Jungblut 2022, https://doi.org/10.1016/j.jplph.2022.153692
L103-105:
The sentences can be extended by adding example species, such as psychrotolerant polar eukaryotic microalga Coccomyxa subellipsoidea c-169 https://link.springer.com/article/10.1186/gb-2012-13-5-r39, and the psychrophilic Antarctic green algae such as ICE-L, ICE-MDV, UWO241 etc. The major difference between psychrophiles and psychrotolerant species is the former can survive at freezing temperatures but will die at more moderate temperatures (Morita, R.Y. (1975). Psychrophilic bacteria. Bacteriol. Rev. 39, 144–167.)
Besides, I only have a couple of minor remarks:
- L17: might consider add word e.g., “relatively” before higher temperatures or simply reuse the word “higher” as “increased”, because we know Antarctica is cold environment for human being so as “lower salinities”
- L154: Table 1 title missing a period at the end.
- L158: Please make sure all species are shorted as the format if had mentioned before, e.g., Navicula criophiliforma as N. criophiliforma, it is OK if they are not shorted in the Figures.
- L289: should label as equation (2)?
- L341: the usage of p-value < 0.05 shall be consistent in the whole article
- L541: might consider aligning the text in centre within the table
- L352: missing period in title of Table 2; the last two columns seem not complete
- Table S1, lacking the table header and maybe consider merging the empty cells or label as NA
Author Response
Dear editor and reviewer 2,
many thanks for your encouraging and constructive comments on our manuscript. We deeply appreciate that you all like the scientific story. We tried our best to address your questions and comments. Below please find our point-to-point responses.
Referee 2:
However, I do have some concerns with regarding to the completeness of background references for broad readers.
L62-64: “while similar studies for Antarctica are missing. ” “As for most other groups of eukaryotic microorganisms remarkably little is still known about marine benthic diatom biodiversity and ecophysiology in Antarctica. “
I think this paragraph shall be extended with more relevant and recent references.
I was surprised the model polar diatom Fragilariopsis cylindrus (Thomas Mock et.al., Nature, 2017 https://www.nature.com/articles/nature20803) was not mentioned in the article. Please also refer to the latest review paper Diatoms and Their Microbiomes in Complex and Changing Polar Oceans, https://doi.org/10.3389/fmicb.2022.786764.
Also, there are many emerging eukaryotic polar model species in recent years, for example the Chlamydomonas nivalis (Brown et al. 2015 https://www.tandfonline.com/doi/full/10.1657/AAAR0014-071 ), Chlamydomonas sp. ICE-L (Zhang et.al., 2020, DOI: 10.1016/j.cub.2020.06.029); Chlamydomonas sp. UWO241 (Zhang et.al., 2021 https://doi.org/10.1016/j.isci.2021.102084). Some recent reviews have also provided in-depth synthesis of research on photophysiology of polar phototrophic organisms to date, which is for authors’ considerations, such as Hüner et al. (2022) https://doi.org/10.1016/j.jplph.2021.153557 Jungblut 2022, https://doi.org/10.1016/j.jplph.2022.153692
Thanks for this suggestion. We expanded in the introduction the following paragraph as requested:
Remarkably little is still known about marine benthic diatom biodiversity and ecophysiology in Antarctica. In contrast, various microalgae of the Antarctic phytoplankton as well as those as-sociated with sea-ice, with snow fields or inhabiting terrestrial sites are much better studied [13,14,15,16 and references therein]. In addition, these more recent publications applied Next-Generation Sequencing (NGS) technologies which greatly expanded current knowledge by providing fundamental information on the underlying molecular mechanisms of physiological and biochemical adaptations to polar environmental conditions.
L103-105:
The sentences can be extended by adding example species, such as psychrotolerant polar eukaryotic microalga Coccomyxa subellipsoidea c-169 https://link.springer.com/article/10.1186/gb-2012-13-5-r39, and the psychrophilic Antarctic green algae such as ICE-L, ICE-MDV, UWO241 etc. The major difference between psychrophiles and psychrotolerant species is the former can survive at freezing temperatures but will die at more moderate temperatures (Morita, R.Y. (1975). Psychrophilic bacteria. Bacteriol. Rev. 39, 144–167.)
We followed this recommendation:
Psychrophilic and psychrotolerant species can be physiologically distinguished as the former can survive at freezing temperatures but will die at more moderate temperatures [29]. Typical examples are the psychrotolerant green microalga Coccomyxa subellipsoidea C-169, which was isolated from a terrestrial site in Antarctica [30], and the psychrophilic unicellular green alga Chlamydomonas sp. ICE-L that thrives in floating Antarctic sea ice [14]. Psychrophilic traits are exemplarily documented in the Antarctic…………..
Besides, I only have a couple of minor remarks:
L17: might consider add word e.g., “relatively” before higher temperatures or simply reuse the word “higher” as “increased”, because we know Antarctica is cold environment for human being so as “lower salinities”
We changed to “increased”
L154: Table 1 title missing a period at the end.
We provided this information
L158: Please make sure all species are shorted as the format if had mentioned before, e.g., Navicula criophiliforma as N. criophiliforma, it is OK if they are not shorted in the Figures.
We followed this recommendation
L289: should label as equation (2)?
Correct, we changed to (2)
L341: the usage of p-value < 0.05 shall be consistent in the whole article
Yes, we checked for consistency
L541: might consider aligning the text in centre within the table
Yes, we did
L352: missing period in title of Table 2; the last two columns seem not complete
We provided the period as well as the missing accession numbers. The latter will be provided soon and added directly to the proofs.
Table S1, lacking the table header and maybe consider merging the empty cells or label as NA
We improved this table
We hope that the reviewer is satisfied with our revision, and now agrees on acceptance.
Best wishes
Ulf Karsten & co-authors